# Hyperbolic Representation of Lateral Force–Displacement Relationship for Underground Installed Pipe

Richard D. Yovichin III [1], Jai K. Jung [2],[*] and Wei-Ju Lee [3]

1   McMillen Jacobs Associates, Mayfield Heights, OH 44124, USA; yovichin@mcmjac.com
2   Department of Civil & Environmental Engineering, Virginia Military Institute, Lexington, VA 24450, USA
3   The Republic of China Marine Corps, Zuoying, Kaohsiung 813779, Taiwan; wjlmarinecorps@gmail.com
*   Correspondence: jungjk@vmi.edu

**Abstract:** Extreme natural hazards such as earthquakes, landslides, and liquefaction create permanent ground deformation (PGD). With the recognition that PGD often causes the most serious local damage in underground structures such as buried pipelines and mining facilities, research and engineering practices for underground structures have focused on soil–structure interaction under PGD. In this study, an underground pipeline was investigated due to its simple geometry. Geotechnical data collection and analysis were used as a study method. Of key importance is the soil–pipe interaction with respect to PGD below the subsurface. This response is typically highlighted by a force vs. displacement relationship and is primarily a function of soil unit weight, depth from surface to the pipe centerline, and the pipe diameter. The non-linear force vs. displacement relationship for transverse horizontal force on a pipe subjected to lateral ground movement, can be represented by a hyperbola. The nonlinear hyperbola can then be turned into a linear line by transforming the axis. This paper investigates a wide range of soil characteristics and summarizes representative hyperbolic parameters for dry medium, dense, and very dense sand for lateral ground movement. The approach is convenient for modeling the soil–pipe interaction and is critical for addressing the complexities of soil and pipe performance, consistent with real-world soil–pipe behavior. The ideas and data analysis techniques presented in this study may be fine-tuned and applied to more complex problems including mining and could ultimately contribute to the management of geotechnical risks.

**Keywords:** soil–pipe interaction; hyperbolic response; lateral pipe movement

## 1. Introduction

Earthquake-induced permanent ground deformation (PGD) often involves large, irrecoverable soil distortion with geometric soil mass changes and large plastic underground structure deformation, involving both material and geometric nonlinearities [1,2]. The effects of PGD not only apply to earthquakes, but also occur in response to floods, landslides, tunneling, deep excavations, and subsidence caused by dewatering or the withdrawal of minerals and fluids during mining and oil production. Such loaded conditions are becoming more important as the concern for public safety increases regarding natural hazards, human threats, and construction in a congested urban environment. These behaviors impose significant demands on the modeling of the soil–structure interaction. Research and engineering practices for an earthquake response for underground structures have focused on PGD and transient ground deformation effects, with the recognition that PGD often causes the most serious local damage in buried structures [3–6]. In this study, an underground pipeline under PGD was investigated due to its simple geometry. However, the ideas and geotechnical data collection/analysis techniques used in this study may help others investigate other soil–structure applications such as mining and limit the potential risk of geotechnical failures.

The soil–pipe interaction under PGD is often performed with one-dimensional finite element models to represent the pipeline and soil force vs. displacement relationships

that are mobilized by various types of ground movement. As described by several design guidelines and previous researchers [7–11], soil–pipeline interaction is represented by components in the axial, transverse, and vertical bearing directions, as represented by the soil springs [1,12,13]. This approach benefits from ease of application and its incorporation into available finite element codes [8] but suffers from the uncoupled representation of soil as a series of spring-slider reactions [14–16]. Many researchers have developed nonlinear soil–pipe force vs. displacement relationships more realistically, using advanced numerical analysis [5,17–21], but they require significant computing power and an in-depth understanding of numerical modeling. The nonlinear behavior of force vs. displacement relationships can be represented by a hyperbola [8,11,14,22]. Such hyperbolas can then be transformed into a linear representation, which makes an analysis of soil–pipe interaction much easier as illustrated in Figure 1. In the figure, F is the measured lateral pipe force, $\gamma_d$ is the dry unit weight of the soil, $H_c$ is the depth from the top of the soil to the center of the pipe, D is the external diameter of the pipe, L is the length of the pipe, Y is a relative displacement of the pipe, and A and B are the hyperbolic parameters. $Y'$ and $Y''$ in Figure 1b are defined in Equations (3) and (7), respectively, and $F'$ and $F''$ are defined in Equations (4) and (8), respectively.

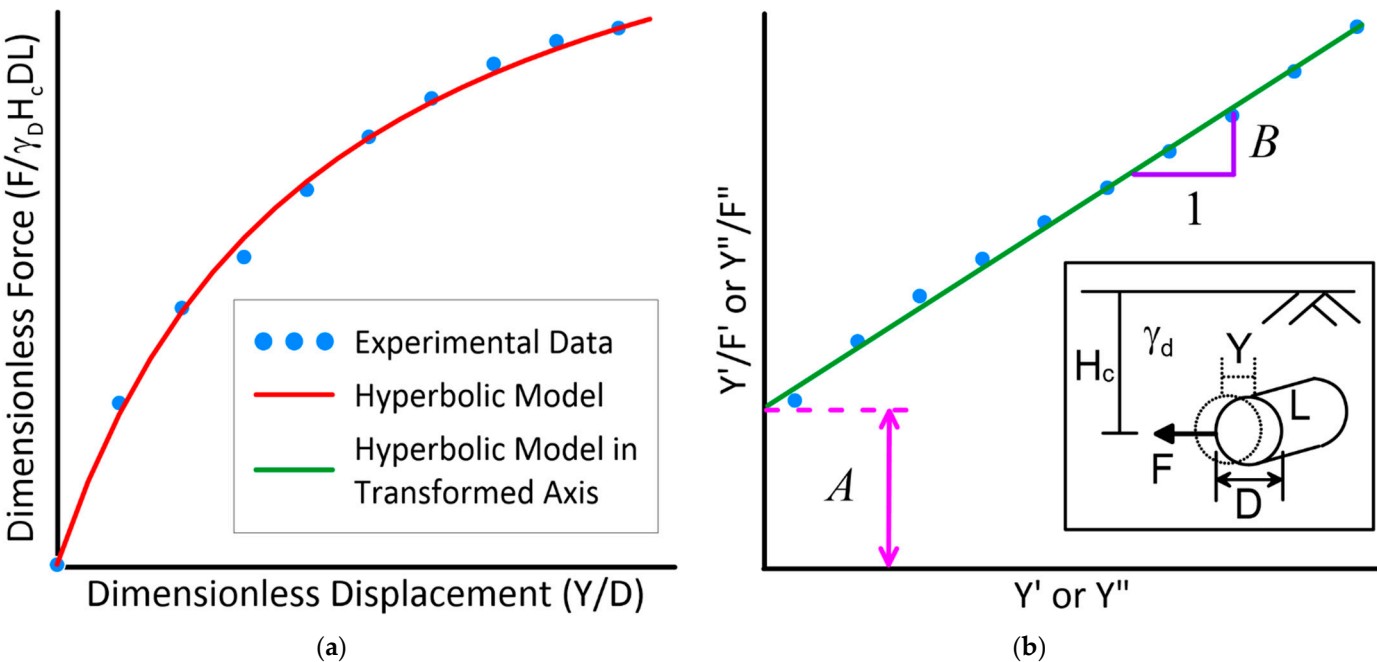

**Figure 1.** Diagram of transformation of (**a**) Nonlinear dimensionless force vs. displacement to a (**b**) Linear representation emphasizing A and B parameters.

The overall objective of this paper is to represent the nonlinear soil–pipe interaction using the hyperbolic parameters *A* and *B* as shown in Figure 1b. As described by Wong and Duncan [23], the *A* and *B* parameters are related to the line of best fit when describing the hyperbolic curve. This will be done by transforming large-scale experimental data into a dimensionless form and using a normalization process plotted on a transformed axis. Providing a simplified hyperbolic method to represent soil–pipe behavior can allow for a user-friendly and relatively quick process based on numerous experimental data as an alternative to advanced numerical analysis.

## 2. Experimental Literature Review

In this study, lateral pipe movement in dry sand is considered and the soil is categorized as medium ($16 \leq \gamma_d \leq 16.7 \text{ kN/m}^3$), dense ($16.7 < \gamma_d \leq 17.5 \text{ kN/m}^3$), and very dense sand ($\gamma_d > 17.5 \text{ kN/m}^3$). In addition, the scope of this research is on the force exerted on the pipe due to the relative displacement between the soil and the pipe. Therefore, the

pipes used in the experiments are very rigid compared to the adjacent soil. Trautmann and O'Rourke [7,9] conducted tests on lateral loaded pipes with ranging overburden ratios from 2 to 11 and pipe diameters between 102 mm and 324 mm. The scope of their research was to investigate how the depth and unit weight of soils effect a force vs. displacement curve for a laterally loaded pipe. The results from their experimentation for medium and dense sand were compared with an analytical model developed by Rowe and Davis [24] and, more recently, by Jung et al. [15] with very good agreement. Based on the test results, Trautmann and O'Rourke [7] derived hyperbolic curves for medium and very dense sand.

Hsu [25] performed a series of tests on two soil types with overburden ratios ranging from 1.5 to 20.5 and pipe diameters between 38.1 mm and 76.2 mm. Hsu's [25] findings for the maximum soil loads were between the results found by Trautmann and O'Rourke [7] and Audibert and Nymann [26]. Hsu [25] investigated the relationship between strain rate and maximum soil loads on the pipe. This allowed Hsu [25] to develop a power–law relationship between the two. By using different strain rates found from the soil loads and pullout rates, Hsu [25] developed a series of rectangular hyperbolic relationships.

Turner [27] performed experiments using glacio-fluvial well-graded sand (referred to as RMS graded sand) for dry and partially-saturated conditions. The density of the dry soil ranged from 16.9 kN/m$^3$ to 17.2 kN/m$^3$, which was checked by a nuclear density gage. The results from testing were slightly higher than Trautmann and O'Rourke [7] and provided modifications to the charts presented by Trautmann and O'Rourke [7]. In addition, Turner [27] provided force vs. displacement curves for soils with 4~10% moisture content and concluded that moist soil loads can be as high as double when compared to dry sand.

Karimian [28] developed the instrumentation capable of recording normal stresses, pullout resistance, displacement of geosynthetics, pipes, and sand grains during testing, and deformation occurring at the surface. Axial and lateral pullout tests were performed on trenched backfill and geotextile-lined trenches, while developing a model for comparison. Karimian [28] found that geosynthetic-wrapped pipes decreased axial soil loads and mentioned that the lateral loading tests were consistent with the rectangular hyperbola reported by Das and Seeley [29] and described by Trautmann and O'Rourke [7]. Karimian [28] suggested that the findings described by Trautmann and O'Rourke [7] and Turner [27] were slightly higher when predicting soil loads on pipes. The developed model was created using a modified hyperbolic model for transverse ground movement. The trenched geotextiles analytical model soil loads were slightly higher than the tests performed, and Karimian [28] stated that this occurrence could be due to localized shear failure.

Olson [30] performed several large-scale lateral pipe movement tests. In general, the dimension of the test box greater than 10 times the pipe diameter is recommended to eliminate the boundary effect [30] and is considered a large-scale test. The purpose of his research was to understand the factors influencing soil performance in soil–pipe interaction, and how to improve accuracy of large-scale testing. The nuclear gage and density scope were used to measure a unit weight of dry and partially saturated RMS graded sand. The comparison of the unit weight, using two different methods, agrees. Olson [30], however, suggested using the nuclear gage for large-scale tests because it is easy to use and a relatively quick process. Using tactile pressure sensors, Olson [30] accurately measured normal stress on the pipe and reported several force vs. displacement curves.

Robert et al. [31] performed two separate tests, under both dry and unsaturated conditions, with two types of soils. Fine Chiba sand [32–34] and coarser RMS graded sands [3,15] were used in these tests. The results from the tests were compared with the numerical simulation. The peak load for unsaturated Chiba sand experiments was greater than the load for dry sand. The RMS graded sand contained nearly the same peak load for unsaturated as for the dry sand conditions; however, the peak load for the unsaturated condition was 10% higher. The unsaturated soil model found similar results for both cases where the pre-failure stiffness of the unsaturated condition was greater than the dry

condition, which agrees with Jung et al. [15]. The types of soils used were found to affect the mechanical behavior and were more apparent in the finer Chiba sand.

Minh and Zhang [18] performed five tests under the monotonic and cyclic forces. The British Standard 137 BSI [35] was used to prepare the very dense sand with $H_c/D$ ratio of 4.77. The diameter and the length of the pipe were 0.351 m and 1.2 m, respectively. The test results show that the soil–pipe behavior basically follows the monotonic rule. They also found that the ultimate soil resistance captured from a monotonic load may differ from that captured from a cyclic load. Of the five tests, only the test performed with the monotonic load was used in this study.

## 3. Experimental Data

### 3.1. Medium Sand Experimental Data

The dry unit weight ($\gamma_d$) of medium sand was categorized as ranging from 16 kN/m$^3$ to 16.7 kN/m$^3$. The tests performed by Trautmann and O'Rourke [7], Karimian [28], Li [36], and Robert [5] fall in this category. The raw experimental data were acquired from Trautmann and O'Rourke [7] and Li [36]. However, the original experimental data were not available from Karimian [28] and Robert [5], and therefore, their data were digitized from their published work. Data shown in Figure 2 encompass these researchers' findings, totaling nineteen experiments. Table 1 describes the specific parameters from each researcher used in the construction of Figure 2.

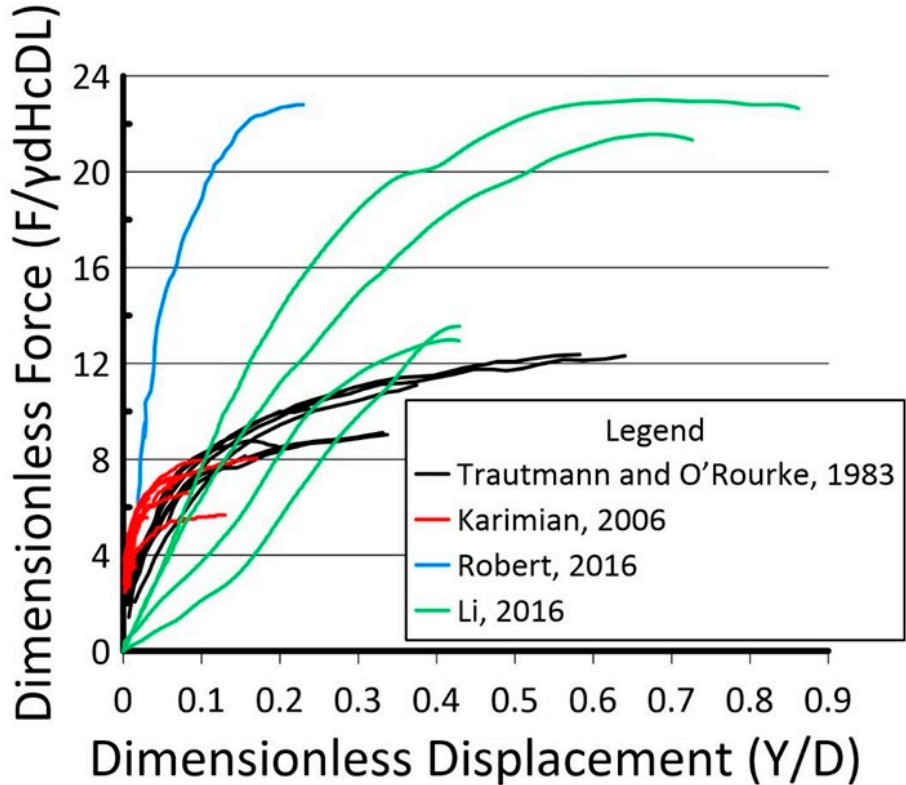

**Figure 2.** Dimensionless force vs. displacement test data for medium sand (16 kN/m$^3$ < $\gamma_d$ ≤ 16.7 kN/m$^3$). Experiments performed by [5,7,28,36].

### 3.2. Dense Sand Experimental Data

The dense sand is defined as the soil dry unit weight from 16.7 kN/m$^3$ to 17.5 kN/m$^3$. Figure 3 shows dimensionless force vs. displacement relationship for lateral pipe movement and is composed of data from four research groups: Hsu [25], Turner [27], Olson [30], and Li [36]. This graph was created using eighteen experiments from the various researchers mentioned above. The original test data were not available for Hsu [25], and the data

were digitized from the experimental data presented by Hsu [25], which were overlapping at various points. This issue caused some difficulty in the data collection process. For Turner [27], Olson [30], and Li [36], raw test data were available and included in the figure. The majority of the data presented in Figure 3 follow a general trend, whereas Li's data [36] have a higher maximum dimensionless force and soft pre-peak behavior when compared to the other researchers' data. Li [36] stated that the compaction while preparing the test bed was not adequately performed and may have affected the test results. Detailed test parameters for dense sand are summarized in Table 2.

**Table 1.** The parameters used for each medium sand test with the corresponding researcher.

| $\gamma_d$ (kN/m³) | Length (mm) | Diameter (mm) | $H_c$/D | Experiment | Note |
|---|---|---|---|---|---|
| 16 | 2400 | 457 | 1.92 | Karimian [28] | 2 tests |
| 16 | 2400 | 324 | 1.92 | Karimian [28] | 3 tests |
| 16 | 2400 | 324 | 2.75 | Karimian [28] | |
| 16 | 1000 | 60 | 3 | Li [36] | |
| 16 | 1000 | 60 | 5 | Li [36] | |
| 16 | 1000 | 60 | 8 | Li [36] | |
| 16 | 1000 | 60 | 10 | Li [36] | |
| 16.4 | 1200 | 102 | 3.5 | T & O [1] [7] | 3 tests |
| 16.4 | 1200 | 102 | 3.5 | T & O [7] | |
| 16.4 | 1200 | 102 | 5.5 | T & O [7] | |
| 16.4 | 1200 | 102 | 8 | T & O [7] | |
| 16.4 | 1200 | 102 | 11 | T & O [7] | 2 tests |
| 16.6 | 2400 | 114.6 | 6 | Robert [5] | |

[1] Trautmann and O'Rourke.

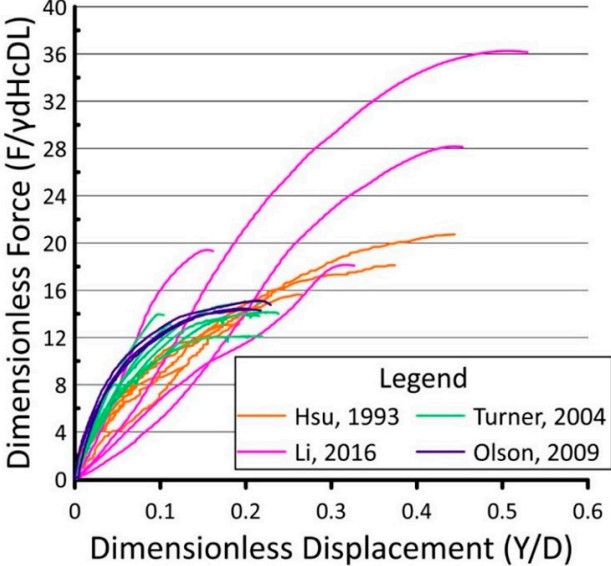

**Figure 3.** Dimensionless force vs. displacement test data for dense sand (16.7 kN/m³ < $\gamma_d \leq$ 17.5 kN/m³). Experiments performed by [25,27,30,36].

### 3.3. Very Dense Sand Experimental Data

Very dense sands are described as having a $\gamma_d$ > 17.5 kN/m³. Trautmann and O'Rourke [7], and Minh and Zhang [18] performed lateral pipe movement experiments on very dense sand. The test results are displayed in Figure 4. Please note that the data shown for Trautmann and O'Rourke [7] in Figure 4 does not represent the entire test recorded by the original researchers. For dense to very dense sands, the hyperbola characterizes the actual curve for displacement less than $Y'_{max}$. A limiting force of $F'_{max}$ is used for displacement greater than $Y'_{max}$. Therefore, the data points exceeding the maximum di-

mensionless force are not shown in the figure. The test data from Minh and Zhang [18] as well as $H_c/D = 1.5$ from Trautmann and O'Rourke [7] were digitized. Table 3 shows the different variables for each test.

**Table 2.** The parameters used for each dense sand test with the corresponding researcher.

| $\gamma_d$ (kN/m$^3$) | Length (mm) | Diameter (mm) | $H_c/D$ | Experiment | Note |
|---|---|---|---|---|---|
| 16.9 | 2440 | 120 | 5.47 | Olson [30] | |
| 16.9 | 1200 | 120 | 5.5 | Turner [27] | |
| 17 | 1000 | 60 | 3 | Li [36] | |
| 17 | 1000 | 60 | 5 | Li [36] | |
| 17 | 1000 | 60 | 8 | Li [36] | |
| 17 | 1000 | 60 | 10 | Li [36] | |
| 17 | 1200 | 120 | 5.5 | Turner [27] | |
| 17.1 | 2440 | 124 | 5.29 | Olson [30] | |
| 17.1 | 1210 | 120 | 5.5 | Turner [27] | |
| 17.2 | 1200 | 76.2 | 2.5 | Hsu [25] | |
| 17.2 | 1200 | 76.2 | 4.5 | Hsu [25] | |
| 17.2 | 1200 | 76.2 | 6.5 | Hsu [25]) | |
| 17.2 | 1200 | 76.2 | 8.5 | Hsu [25] | |
| 17.2 | 1200 | 76.2 | 10.5 | Hsu [25] | |
| 17.2 | 2440 | 124 | 5.29 | Olson [30] | |
| 17.2 | 1200 | 120 | 5.5 | Turner [27] | 3 tests |

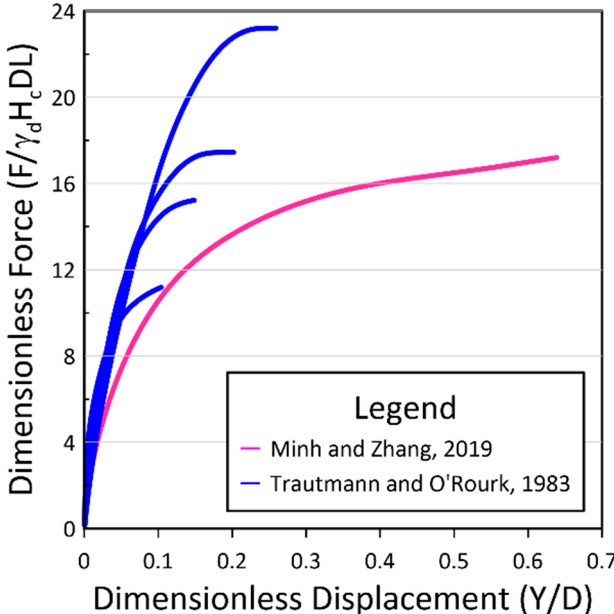

**Figure 4.** Dimensionless force vs. displacement test data for dense sand (16.7 kN/m$^3$ < $\gamma_d$ ≤ 17.5 kN/m$^3$). Experiments performed by [7,18].

**Table 3.** The parameters used for each very dense sand test with the corresponding researcher.

| $\gamma_d$ (kN/m$^3$) | Length (mm) | Diameter (mm) | $H_c/D$ | Experiment |
|---|---|---|---|---|
| 17.7 | 1200 | 120 | 3.5 | T & O [1] [7] |
| 17.7 | 1200 | 120 | 5.5 | T & O [7] |
| 17.7 | 1200 | 120 | 8 | T & O [7] |
| 17.7 | 1200 | 120 | 11 | T & O [7] |
| 20.9 | 1200 | 351 | 4.8 | Minh and Zhang [18] |

[1] Trautmann and O'Rourke.

## 4. Data Analysis

To begin performing the data analysis, an intermediate step is needed. This step involves finding the maximum force ($F_{max}$) with the corresponding maximum displacement ($Y_{max}$), as described by Jung et al. [1]. The maximum dimensionless force is selected where a clear peak can be seen in the force vs. displacement graph or in a dimensionless graph as shown in Figure 5b. However, in some cases, especially for medium sand, a definitive peak is not visible. In such cases, a methodology as illustrated in Figure 5a is used to define $F_{max}$. In the figure, $(F)_{ult}$ is the asymptotic value of the principal force, which the force vs. displacement curve approaches at the infinite displacement. As shown in the figure, the hyperbola remains below $(F)_{ult}$ within all finite values of displacement. The force at the maximum lateral pipe force, $F_{max}$, is expressed as:

$$F_{max} = R_f \times (F)_{ult} \tag{1}$$

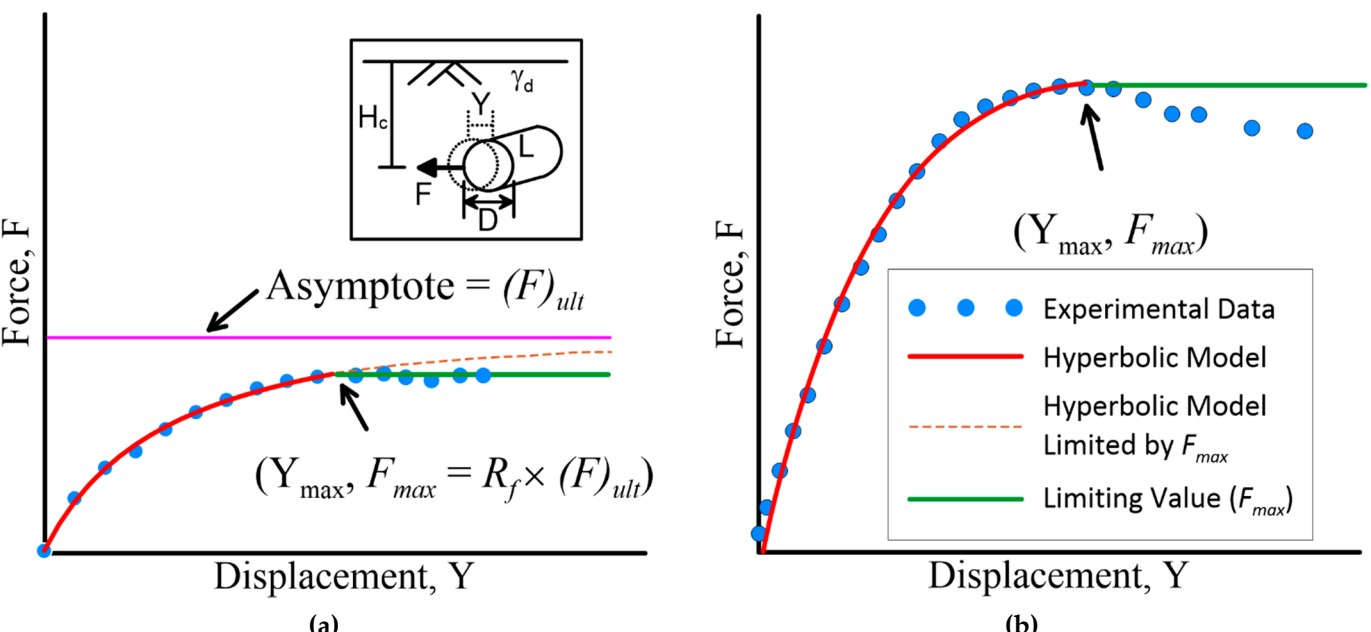

**Figure 5.** Determining $F_{max}$ and $Y_{max}$ from experimental data for (**a**) when a clear peak is not visible and (**b**) when a clear peak is visible (Reprinted with permission from Ref. [7]. Copyright 1983 ProQuest).

Duncan [37] explains how the over estimation of maximum force is typically around 11% on average, therefore, a reduction factor ($R_f$) is needed to correct the over estimation. Wang and Duncan [23] reported that $R_f$ typically ranged between 0.5 and 0.9 for most soils. They also reported that the compressive strength of the soils is always less than $(F)_{ult}$. In the analysis, $F_{max}$ is defined as $F_{max} = R_f (F)_{ult}$, where $R_f = 0.9$, which is in agreement with the information used by Trautmann and O'Rourke [7,9] and falls in the range of $R_f$ reported by Wang and Duncan [23]. This approach is also used by Yimsiri et al. [17] and Jung et al. [15,22], which was developed from the method discussed and used by several researchers [7,37–39]. When a clear peak was not visible, a hyperbolic curve was fitted from the obtained force vs. displacement data by means of maximum force extrapolation, using a similar approach noted by Trautmann and O'Rourke [7] for characterizing the force vs. displacement relationship. In the analysis, the following hyperbola was fitted from the force vs. displacement relationship:

$$F = \frac{Y}{\alpha + \beta Y} \tag{2}$$

where $1/\alpha = \lim_{Y \to 0} (dF/dY)$ corresponds to initial force vs. displacement stiffness curve and $1/\beta = \lim_{Y \to \infty} (F)$ corresponds to $(F)_{ult}$ [17]. By extrapolation, the hyperbolic curve was followed to find the location of $(F)_{ult}$ or the asymptote. $(F)_{ult}$ was then multiplied by a reduction factor of 0.9. The value found from this multiplication process is considered $F_{max}$. Yovichin [40] summarized and reported the corresponding values of each test. Once $F_{max}$ and $Y_{max}$ are defined for each test, the normalization process of dimensionless force vs. dimensionless displacement could occur. The normalization process allowed for each soil type to be presented in two different ways, as discussed in the next section.

## 5. Methodology

### 5.1. Single-Normalization

The hyperbolic stress-strain relationship was first developed by Kondner [38] and further discussed and used by Wang and Duncan [23] and Trautmann and O'Rourke [7]. First, the dimensionless force vs. displacement was fitted and plotted on a transformed axis, $Y'/F'$ and $Y''$ where

$$Y' = \frac{Y}{D} \tag{3}$$

$$F' = \frac{F}{\gamma_d H_c D L} \tag{4}$$

Each parameter used in Equations (3) and (4) are defined previously. Plotting the dimensionless force vs. displacement on a transformed axis allows the hyperbolic curve to be represented by a linear line following the rectangular hyperbola:

$$F' = \frac{Y'}{A + BY'} \tag{5}$$

The parameters describing the hyperbolic equation have physical significance and are shown in Figure 1b, labeled as $A$ and $B$. The $A$ parameter is the reciprocal of the initial tangential modulus, whereas the $B$ parameter is the reciprocal of the asymptotic value of $F_{ult}$, as discussed in the previous section.

The single-normalization process allows the data to be represented by a linear line on the transformed axis plot. This linear line is then fitted with a linear best-fit line. Furthermore, the best-fit line will represent the best-fit hyperbola [23]. Similar graphs shown in Figure 1b are generated for each test's data, by plotting the dimensionless force and displacement on a transformed axis. The $A$ and $B$ parameters for each test previously mentioned, are then found and recorded.

### 5.2. Double-Normalization

After single-normalization occurred, the hyperbolic curve is then converted to a double-normalized force vs. displacement graph, by plotting on a transformed axis $Y''/F''$ and $Y''$ [38] (Figure 1b) as

$$F'' = \frac{F'}{F'_{max}} \tag{6}$$

$$Y'' = \frac{Y'}{Y'_{max}} \tag{7}$$

where $F'_{max}$ and $Y'_{max}$ are the maximum values of $F'$ and $Y'$, respectively. The double-normalized force vs. displacement graph can then be approximated by Equation (8).

$$F'' = \frac{Y''}{A + BY''} \tag{8}$$

The average value of $Y'_{max}$ is 0.054 $H_c$, 0.036 $H_c$, and 0.025 $H_c$ for medium, dense, and very dense sand, respectively. The double-normalization process followed a similar approach in finding the $A$ and $B$ parameters, as described in single-normalization

(Equation (5)). Equations (3) and (4) are similar to Equations (6) and (7), with the maximum values, $F'_{max}$ and $Y'_{max}$, playing an influential role in the double-normalization process, whereas single-normalization does not take into account the maximum force or displacement. The other researchers [9,41] also focused their attention on double-normalization when reporting the soil–pipe force vs. displacement relationship.

*5.3. Outliers*

After the *A* and *B* parameters are generated for each data set, the outliers are identified. The *A* and *B* parameters for single- and double-normalization for each experiment, based on soil type, are input individually and a box plot is generated. A box and whisker plot is a graphical representation of the data as shown in Figure 6. The distance between the upper (Q3) and lower quartiles (Q1) is the inter quartile range (IQR). This range is classified as 50% of the results. The minimum and maximum values are considered the upper and lower whiskers; however, if these values exceed 1.5 IQR they are deemed outliers. These outliers are further classified as mild or extreme, >1.5 IQR and >3.0 IQR, respectively [42,43]. In this study, the mild classification is used and the *A* or *B* parameters greater than 1.5 IQR are neglected for both normalizations. The outlier analysis is useful to find the representative *A* and *B* values for each soil type.

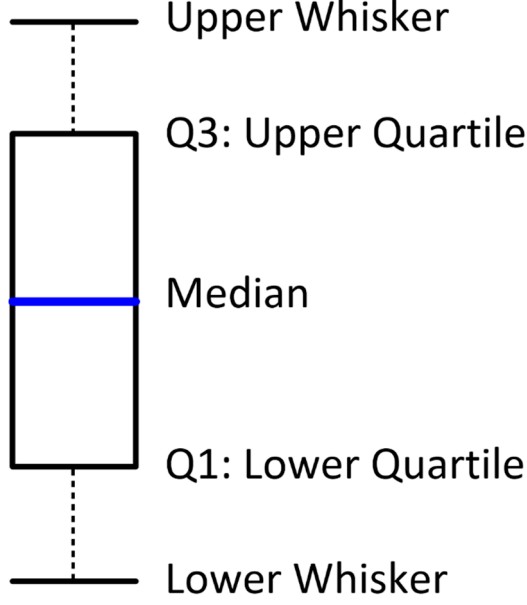

**Figure 6.** Box and Whisker Plot.

It turned out that there were no outliners for very dense sand. For medium sand, five tests performed by Robert [5] and Li [36] were found to be outliers. For dense sand, out of a total of eighteen tests from four researchers [25,26,30,36], four tests performed by Li [36] were found to be outliers.

## 6. Results and Discussion

Excluding the outliers, the single- and double-normalized graphs for medium sand are generated and displayed in Figure 7a,b, respectively. The single-normalization appears to show the data beginning at the origin. This is due to the values of the y-intercept (*A* parameter) being relatively small. The double-normalization graph, however, allows each experiment to be more visible with less congestion, causing a larger range for the *A* parameter to be produced. All the experiments shown in the figure exhibit a linear trend. By the nature of each normalization process, the maximum $Y'/F'$ and maximum $Y'$ varies for each individual experiment in the single-normalized graph, whereas the double-normalization process forces the maximum $Y''/F''$ and maximum $Y''$ to be 1.

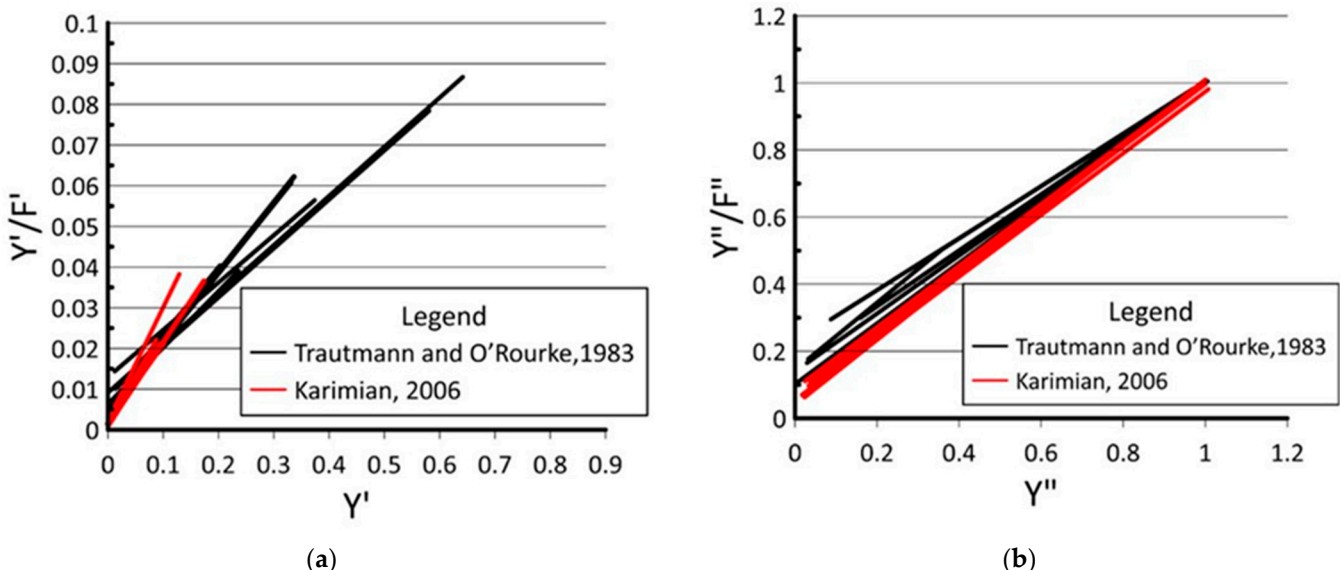

**Figure 7.** Diagram of (**a**) Single- and (**b**) Double-normalization for medium sand. Experiments performed by [7,28].

The *A* parameter for single-normalization obtained from Karimian [28] ranges from 0.0005 to 0.0014 with a $\gamma_d$ = 16 kN/m$^3$ and H$_c$/D varying from 1.92 to 2.75. The *B* parameter from Karimian [28] bound between 0.07 and 0.17. Trautmann and O'Rourke [7] had a slightly higher *A* parameter ($0.002 \leq A \leq 0.0074$) and a smaller *B* parameter ($0.07 \leq B \leq 0.11$); however, the $\gamma_d$ was 16.4 kN/m$^3$ and the H$_c$/D ranged from 3.5 to 11. When comparing the two researchers' data separately, the *A* and *B* parameters for single-normalization did not overlap, yet this occurrence did not happen with the double-normalization case.

The double-normalization process allowed for the data from each researcher, regardless of the independent variable (e.g., $\gamma_d$, H$_c$/D), to be well converged. For example, the *A* parameter for Trautmann and O'Rourke [7] is distributed between 0.06 and 0.22, and the *B* between 0.78 and 0.96. Karimian's [28] ranges are similar to Trautmann and O'Rourke [7], from 0.05 to 0.10 and from 0.89 to 0.95 for *A* and *B* parameters, respectively. The double-normalization process allows each respective parameter to correspond to the other test results. The maximum and minimum values of *A* and *B* parameters for medium sand are summarized in Table 4. The single-normalized *A* parameters are less than the double-normalized, and this is also the case for the *B* parameters. These differences in the *A* and *B* parameters are inherently visible in Figure 7. Detailed *A* and *B* parameters for each test are reported in Yovichin [40].

**Table 4.** Maximum and minimum values of medium sand.

| Parameter | Single-Normalized | | Double-Normalized | |
|---|---|---|---|---|
| | *A* | *B* | *A* | *B* |
| **Maximum** | 0.0074 | 0.17 | 0.22 | 0.96 |
| **Minimum** | 0.0005 | 0.07 | 0.05 | 0.78 |

The results for dense sand are visualized in Figure 8 and the maximum/minimum values of the *A* and *B* parameters are summarized in Table 5. The H$_c$/D for each experiment varied from 2.5 to 10.5. Both the single- and double-normalization graphs, plotted on a transformed axis, display a linear trend for most of the experimental data. For single-normalization, the *A* parameter was approximately between 0.0024 and 0.0073, and the *B* parameter was between 0.03 and 0.06. The double-normalization parameters have a larger range, with *A* being between 0.16 and 0.57 and *B* between 0.38 and 0.81.

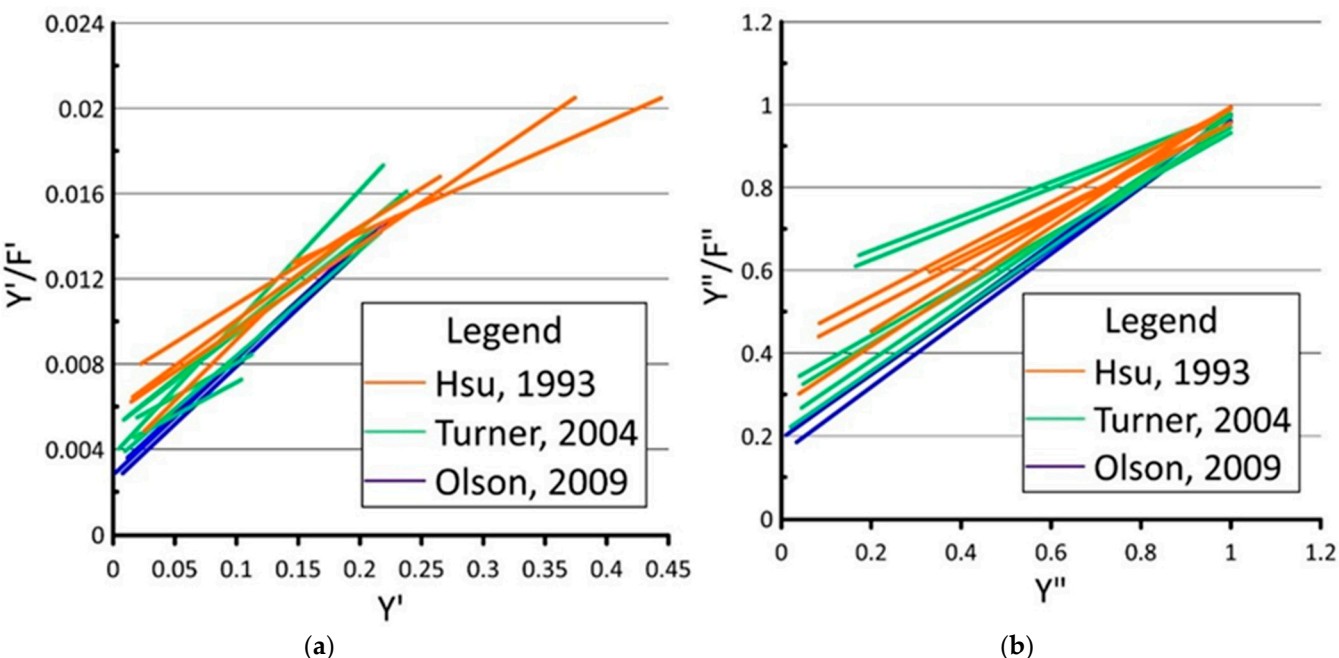

**Figure 8.** Diagram of (**a**) Single- and (**b**) Double-normalization for dense sand. Experiments performed by [25,27,30].

**Table 5.** Maximum and minimum values of dense sand.

| Parameter | Single-Normalized | | Double-Normalized | |
|---|---|---|---|---|
| | *A* | *B* | *A* | *B* |
| **Maximum** | 0.0073 | 0.06 | 0.57 | 0.81 |
| **Minimum** | 0.0024 | 0.03 | 0.16 | 0.38 |

Tests from Trautmann and O'Rourke [7] and Minh and Zhang [18], classified as very dense sand, are shown in Figure 9. The single-normalization graph is spread out until a dimensionless displacement ≤0.075. In the double-normalization, the test data converge into a single linear line. The $H_c/D$ varied from 1.5 to 11. It is found that as $H_c/D$ increases, the *A* parameter increases, whereas the *B* parameter decreases for both single- and double-normalization. Table 6 below illustrates the ranges of the *A* and *B* parameters, regardless of the $H_c/D$. The single-normalization *A* parameters are between 0.0008 and 0.0038, whereas for double-normalization *A* flocculates from 0.10 to 0.30. The *B* parameters are from 0.03 to 0.11, and 0.65 to 0.91 for single and double, respectively.

**Table 6.** Maximum and minimum values of very dense sand.

| Parameter | Single-Normalized | | Double-Normalized | |
|---|---|---|---|---|
| | *A* | *B* | *A* | *B* |
| **Maximum** | 0.0038 | 0.11 | 0.30 | 0.91 |
| **Minimum** | 0.0008 | 0.03 | 0.10 | 0.65 |

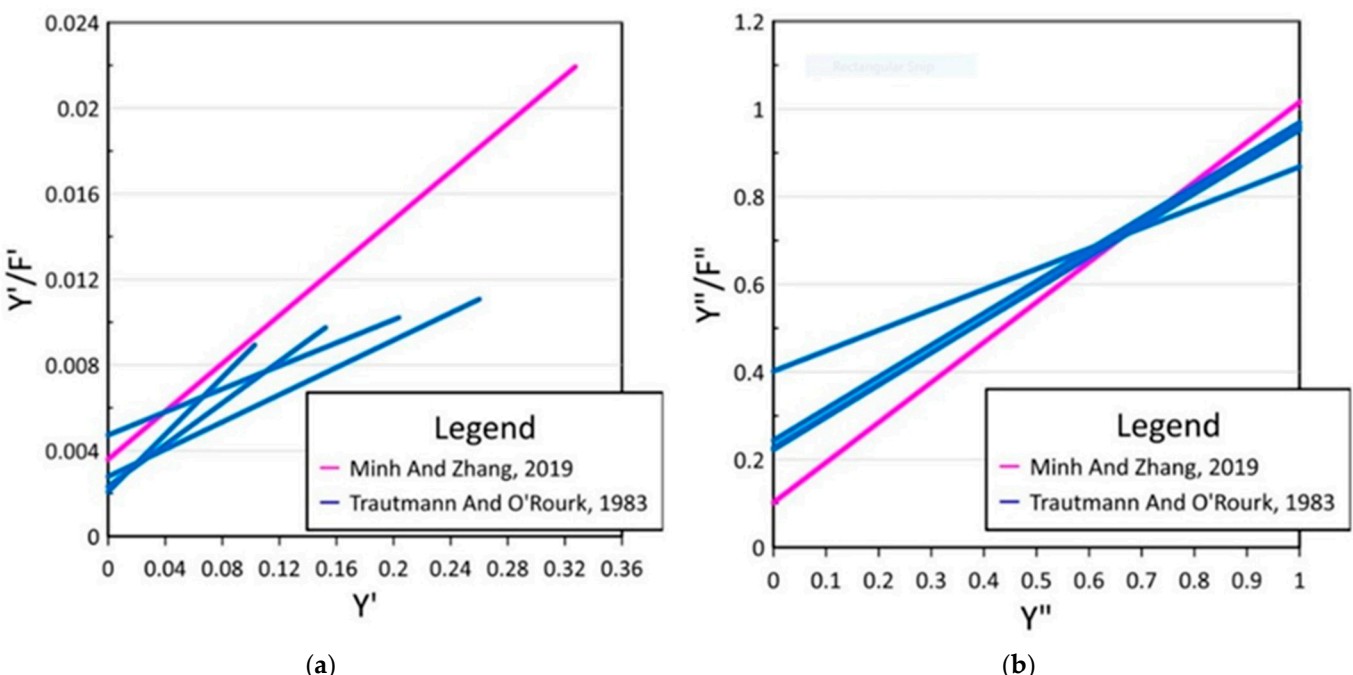

**Figure 9.** Diagram of (**a**) Single- and (**b**) Double-normalization for very dense sand. Experiments performed by [7,18].

### 7. Compiled Data and Validation

In general, medium sand has a higher *B* parameter when compared to the other soils. As unit weight increases, the *B* parameter decreases; however, the *A* parameter remains relatively the same. The correlation between *A* and *B* changed significantly with respect to double-normalization. The *A* and *B* parameters for double-normalization, plotted on the x- and y-axis, respectively, are shown in Figure 10. All the data points found exhibit a linear trend, even though the unit weight varied for each experiment. A linear regression line with a high $R^2$ correlates the *A* and *B* parameters with the negative sloped line as shown in Figure 10. The data points in dense sand have the highest range (distance between maximum and minimum) of *A* and *B* parameters. The medium sand *B*-parameter range was less than very dense sands, whereas the *A*-parameter range was slightly greater. To the best of the authors' knowledge, the correlation shown in Figure 10 between the *A* and *B* parameters has not been reported elsewhere.

The average *A* parameter for single-normalization is converged to 0.003, regardless of soil type. The average *B* parameters for medium, dense, and very dense sand are 0.11, 0.04, and 0.06, respectively. With respect to double-normalization, the *A* parameters for medium, dense, and very dense sand are 0.10, 0.33, and 0.15, respectively, and *B* parameters are 0.91, 0.64, and 0.84, respectively. For medium sand, Trautmann and O'Rourke [7] reported the *A* and *B* parameters for double-normalization as 0.1 and 0.9, respectively, which are in agreement with the values found in this research. Trautmann and O'Rourke [7], however, did not mention parameters for dense soils. The results reported by Trautmann and O'Rourke [7] for very dense sand were 0.25 and 0.75 for *A* and *B* parameters, respectively. In this study, the *A* and *B* parameters for very dense sand are slightly different with 0.21 and 0.76, respectively.

To validate Figure 10, a double-normalized force vs. displacement curve from an independent test performed by Turner [27] is compared with one using the methodology described above. The soil properties of the test are $H_c/D = 5.5$, $D = 117$ mm, $\gamma_d = 16.7$ kN/m$^3$, and $L = 1200$ mm. The lateral maximum dimensionless force is obtained from Jung et al. [15] and converted to $F'_{max}$. Using the average value of *A* parameters for dense sand (Section 7), the *B* parameter is calculated using Figure 10. Then using Equations (6)–(8) and $Y'_{max}$ as reported

in Section 5.2, the force vs. displacement curve is approximated and plotted in Figure 11. The force vs. displacement curve from Turner [27] is also plotted in Figure 11. As shown in Figure 11, the curve from Turner [27] and that from Equation (8) agree very well. The favorable comparison implies that the correlation shown in Figure 10 is suitable to approximate a force vs. displacement relationship for later pipe movement in dry soil.

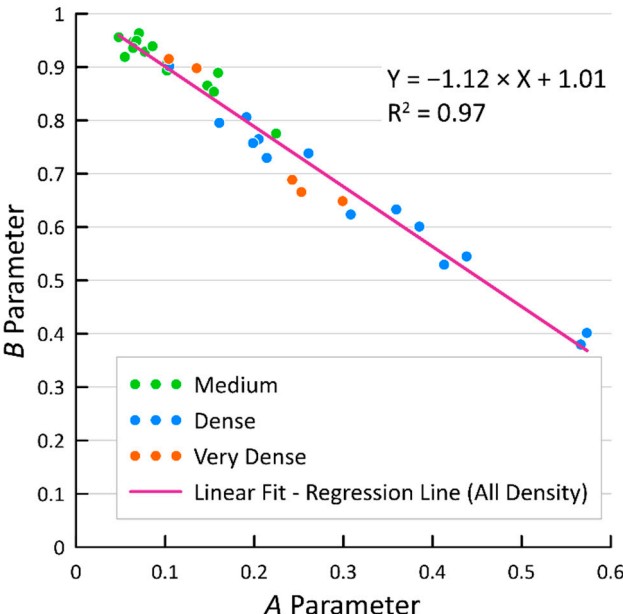

**Figure 10.** Correlation of A and B parameter for double-normalization.

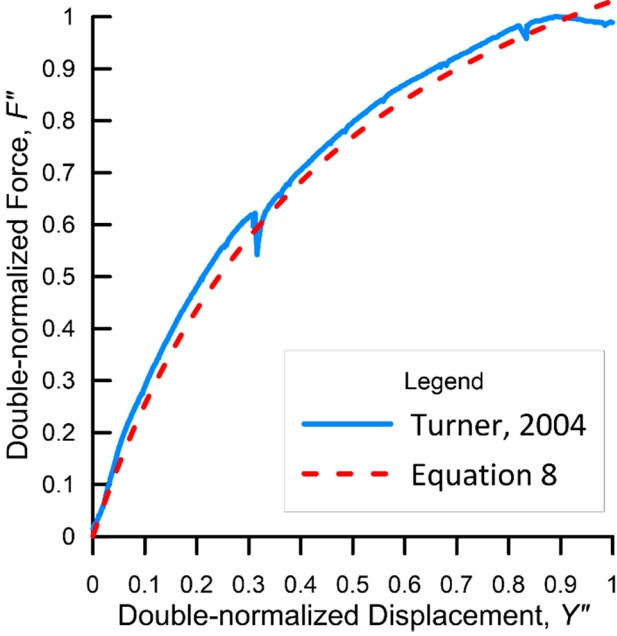

**Figure 11.** Validation of Figure 10 and Equation (8). Experiments performed by [27].

## 8. Conclusions

A force vs. displacement behavior for lateral pipe movement was conducted on three types of soil: medium, dense, and very dense sands. This study focused on experiments performed by several researchers involving $H_c/D \geq 1.5$, where the *A* and *B* parameters for single- and double-normalization plotted on a transformed axis were investigated. An extensive literature review was designated to understand how the tests were conducted,

what types of tests were suitable for this case study, and how the parameters would be found. The minimum and maximum *A* and *B* parameters are reported for all respective soil types. For validation, the results found by Trautmann and O'Rourke [7] regarding *A* and *B* parameters were compared and agree for medium and very dense sand.

An effort was made to obtain a correlation between the *A* and *B* parameters. The correlation was not visible for single-normalization. However, there was a negative linear relationship with a high statistical significance for double-normalization. The variety of tests from different researchers allows for a wide range of soils to be represented. This allows for the *A* and *B* parameters to be generalized. Due to the generalization, this study can help engineers replicate a soil–pipe interaction without a sophisticated modeling technique. The geotechnical data collection and analysis techniques used in this study may be further used in investigating other types of soil–structure application such as mining and could ultimately limit the potential risk of geotechnical failures.

## 9. Limitations of the Study

This report is a systematic review study. The data were collected from other researchers within this field of study. While conducting the literature review, the authors were able to find twenty-eight tests for loose sand having $\gamma_d < 16 \, \text{kN/m}^3$ from several researchers [5,25,44–46]. However, the results showed low R-squared values of *A* and *B* parameters for both single- and double-normalization. Therefore, the methodology described in this paper is not applicable for loose sand. In addition, all pipes in this research have a diameter smaller than 500 mm and the majority are 100~150 mm. Therefore, the method proposed in this paper may not be suitable for pipelines greater than 500 mm in diameter. Additional large-scale testing is recommended to help quantify the 500 mm or larger pipe condition.

**Author Contributions:** R.D.Y.III: data curation, formal analysis, investigation, software, validation, writing—original draft preparation; J.K.J.: Conceptualization, methodology, supervision, validation, writing—review and editing; W.-J.L.: data curation, investigation, software, visualization. All authors have read and agreed to the published version of the manuscript.

**Funding:** This research received no external funding.

**Data Availability Statement:** Not applicable.

**Acknowledgments:** Key Laboratory of Urban Security and Disaster Engineering of Ministry of Education at Beijing University of Technology, Li, and other researchers are acknowledged for their support. The former and current department heads of the Civil and Environmental Engineering at Virginia Military Institute, Riester and Newhouse, are also very much appreciated for their kind support. The first author thanks the Youngstown State University for offering the University Research Council Scholarship, and the other authors thank the Virginia Military Institute for SURI Scholarship support.

**Conflicts of Interest:** The authors declare no conflict of interest.

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
