# Peer review of "Hyperbolic Representation of Lateral Force–Displacement Relationship for Underground Installed Pipe"

_mining, doi:10.3390/mining2020010_

Round 1
Reviewer 1 Report
The paper is well-written and easy to read and follow. However, the research is brief and methodology is very simple (just a regression of previously published results). In my opinion at the current stage the manuscript is not at the standard expected from a journal publication. Can you add a case study and compare results from hyperbolic pipe-soil interaction with proposed linear expression, for different pipe parameters and soil densities?
Reviewer 2 Report
The paper is well-written and I enjoyed reading the paper. Please consider addressing the following minor comments:
- - Please define Y', Y" in Figure 1(b)
- - Although the review of the previous numerical studies is not the scope of this paper, mentioning some of the key studies is always appreciated by the readers.
- - Please add some short discussions on the filtered data sets in Section 3.1.
- - Please add the justifications behind neglecting the data points exceeding the maximum dimensionless force for Trautmann and O’Rourke in Section 3.2.
- - It is not clear from a reader's perspective on how the Fmax is selected for the medium dense sand in Section 4. Please consider rewording this section.
- - Please provide some examples instead of the theory in Section 5.3.
- - Almost 26% and 22% data for medium and dense sand are outliers which might question the methodology.
- - Was there any outlier for very dense sand? Seems like nothing is reported on this issue.
- - Please show an example on how Fig 10 can be used in the practical projects.
- - Previous researchers have shown that the pipe diameter can significantly affect the lateral resistance values. Does the method proposed in this paper take that into account?
- - Please consider adding some references from the numerical studies on the lateral pipe-soil interaction, e.g. Roy et al. (2016), Pike et al. (2016) etc.
Reviewer 3 Report
General comments:
This study examines the method of expressing the lateral force-displacement relationship of a pipe buried in the ground with a hyperbolic representation. The results of experiments conducted by other researchers so far are summarized, and the accuracy of dimensionless force-displacement relations is examined. The authors have compiled many experimental results and examined the data and found the linear relation between the constant parameters of the hyperbolic model.
First, it is necessary to clarify the setting conditions of the target problem. In addition, although the content of the study itself seems to be less novel, it was judged to be practically useful in terms of compiling the experimental data so far. The points that are suspicious in the content, the items that need improvement, and the points that were pointed out in the editorial are listed below.
â– Matters that must be corrected
- Figue1:An additional diagram is needed that illustrates the setting conditions for the problem that the paper is targeting.
・The notation of y should be defined in the figure.
・Where is the displacement Y in the pipe? The displacement of the pipe seems to be a function in the length direction. The displacement Y also depends on the boundary conditions that support the pipe and how the ground displacement is applied, so it is necessary to show the experimental setup.
・Please clarify in the explanatory diagram what kind of experiment the force F is obtained from.
- Table 1-3: The experimental data are influenced by the flexural rigidity and material of the pipe model. These values ​​should be as clear as possible. Since the research is compiling the experimental results, it is necessary to show the experimental conditions in as much detail as possible.
â– Improvement points and editorial point
- Line114: The definition of a large-scale test should be clarified.
- Line192, 194 and 211: “pick” may be “peak” or “convergence value”?
- Legend of Figure 4 : “And” should be “and”
Round 2
Reviewer 1 Report
The authors have addressed my comments.
Reviewer 3 Report
The manuscript has been properly modified.